# Sliding Corrosion Fatigue of Metallic Joint Implants: A Comparative Study of CoCrMo and Ti6Al4V in Simulated Synovial Environments

**Jae Joong Ryu [1,*], Edward Cudjoe [1], Mihir V. Patel [1] and Matt Caputo [2]**

[1] Rayen School of Engineering, Youngstown State University, Youngstown, OH 44555, USA; edwardcudjoe@gmail.com (E.C.); mihirvp12@gmail.com (M.V.P.)

[2] Department of Science, Penn State Shenango, Shenango, PA 16146, USA; mpc215215@psu.edu

* Correspondence: jjryu@ysu.edu

**Abstract:** Mechanical contact in a corrosive synovial environment leads to progressive surface damage at the modular interface of the joint implants. The wear debris and corrosion products degrade the synovial fluids and change the lubrication mechanisms at the joints. Consequently, the unstable joint lubrication and corrosion products will further induce the undesirable performance of the joint implants. In this study, the two major joint materials, CoCrMo and Ti6Al4V, were tested during the course of reciprocal sliding contact in simulated synovial liquids. Open circuit potential and coefficient of friction were monitored to describe electrochemical and mechanical responses. Potentiostatic test results illustrated electrochemical damage on both surfaces that modified oxidation chemistry on both surfaces. However, more significant modification of the CoCrMo surface was detected during wear in the simulated joint liquid. Even with a reduced coefficient of friction on the CoCrMo surface in sodium lactate environments, fretting current density drastically increased in corrosive sodium lactate with pH 2. However, the test results from the Ti6Al4V surface presented less coefficient of friction values, and moderate change in fretting current. Therefore, the experimental study concluded that the biocompatibility of Ti6Al4V is superior to that of CoCrMo in the combined effect of mechanical loadings and an electrochemical environment.

**Keywords:** wear; contact fatigue; synovial friction; corrosion; total joint replacements

## 1. Introduction

Total joint replacements (TJR) are considered to be one of the most successful reconstructive orthopedic surgeries that can be implemented for functional restoration of damaged joints [1–3]. Normal synovial joints provide very smooth motion between limb segments because the articular cartilage and synovial fluid minimize friction at the synovial interface [4–6]. However, total joint replacement removes the natural articular cartilage and replaces it with a synthetic joint material. It inherently modifies the nature of synovial joint lubrication, and inevitably loses the smooth mobility and wear protection of the artificial joints. The aggressive shearing motion during joint articulation generates insidious tribological problems such as surface fatigue and subsequent wear on the modular interfaces of the joints. Although the modular design of joint replacements is mostly due to ease of customization for individuals, modular interface such as the stem-head interface of hip replacements is continually subjected to wear and corrosion. The wear debris migrates locally and systemically, and the corrosion products cause osteolysis in periprosthetic tissue [7,8]. The multifactorial bio-tribological behavior of active articulation denaturalizes the physiology at periprosthesis, and the unhealthy physiopathological condition of the synovial fluid significantly limits the longevity of joint implants [9–11].

Prosthetic implant materials are used to improve mechanical functions for smooth articulation and load bearing. As they perform biomechanical functions in the human

body, local and systemic stability in biological and physiological environments is critical. Such implantable materials are made from three main materials: metals, ceramics and polymers [12,13]. Metal alloys for prosthetic implants have been highly preferable because of mechanical and manufacturing advantages, including less total loss of material being removed, low chance of dislocation, and a reduced chance of fracture [14–16]. However, only a few metal alloys, including titanium based and cobalt-chromium based alloys, are biocompatible and have been successful as orthopedic implants for long-term services.

Titanium (Ti) alloys are the most widely used for medical devices and orthopedic implants because of their excellence of inertness in a biochemical environment, and their superior mechanical strengths [16–18]. In comparison to other alloys such as cobalt chromium and steels, titanium reveals favorable specific strength (strength/density). The alloying elements of titanium considerably impact the transformation temperature between $\alpha$-HCP and $\beta$-BCC, and stabilize the phases [19]. Most commercially available medical-grade titanium alloys contain aluminum and vanadium elements because they are soluble in both $\alpha$ and $\beta$ phases, and particularly because they improve strength and ductility [20–22]. The most common titanium alloys used in making implants are commercially pure titanium (ASTM F67) and extra-low interstitial Ti6Al4V alloy (ASTM F136). Cobalt chromium (CoCr) alloys are being used for the bearing components of joint replacements due to their high stiffness and mechanical strengths with stable chemical responses in the human body. The greater hardness of CoCr alloys also results in excellent wear resistance. Cobalt Chromium alloys exhibit significantly improved strength and chemical inertness through the addition of molybdenum and the limits of other elements (ASTM F75 and F1537) [23,24]; such properties qualify its usefulness in load-bearing joint implant components.

As aforementioned, Ti and CoCr alloys form a stable passive layer that protects the metal substrate from corrosion attack. The mechanical and chemical stability of the passive film is very important, in order to extend the useful life of joint implants. The hard surface layer reduces adhesion between the interacting metal surfaces, and thus it effectively lowers friction force. This passive layer constructs a kinetic barrier of chemical activation, and separates the base metals from their surrounding reactive environments. Therefore, the greater hardness and inertness of the oxide layer results in superior wear and corrosion resistance against the biomechanical stimuli [25,26].

In spite of their excellent mechanical and chemical properties, the course of daily motions of the patient and the corrosive synovial fluids surrounding the implant progressively deteriorate the implant surfaces and lead to unexpected failure of the joint. When frictional contact of joint components damages the surface layer during active articulations, the oxide film loses its protection from corrosion attack [27–30]. Through the damaged surface area, the increased electrochemical potential leads to rapid metal ion dissolution, and reforms the passive layer instantaneously. Corroded wear debris and dissolved metal ions generated by joint friction are usually associated with physiologically adverse relations in the periprosthetic tissues [31,32]. Recent research has proven that high early failure rate of hip implant has been directly related to fretting corrosion of the modular taper interfaces of hip joint replacements. However, due to their superior mechanical and biochemical performance, Ti and CoCr alloys are continually used in current joint implant manufacturing. The previous investigation of tribocorrosion of metallic implants indicates that the CoCr surface showed more wear rate than the Ti alloy surface under the same mechanical and electrochemical stimuli, despite the greater hardness of the Co-Cr surface compared to that of the Ti alloy surface [33–36]. Previous studies agreed with the retrieval studies of hip replacements: at the CoCr–Ti junction, the harder CoCr surface showed faster wear behavior than the counter Ti surface [12,37]. This is because the combined complexity modifies the damage mechanisms when the mechanical damage process takes place in chemically reactive environments. The synergism of mechanical loading and a reactive synovial environment alters tribocorrosion behaviors, and ultimately its effect differs in damage processes on Ti and CoCr alloys. CoCr alloys manifest superior strength and hardness compared to Ti alloys. Therefore, it has been understood that the longevity of CoCr joints is greater than that

of Ti joints. However, previous clinical studies and in vitro simulations of tribocorrosion presented contrary results, thus the CoCr implant surface showed higher wear rate than that of the Ti implant surface despite CoCr's superior mechanical properties [12,15].

In this paper, CoCrMo (F1537) and Ti6Al4V (F136) alloys were subjected to continuous sliding contact in a corrosive environment. In order to understand the chemo-mechanical synergism and damage recovery behaviors of the two different oxides, fretting contact experiments were carried out in a tribochemical cell. The electrochemical and mechanical evolutions of the alloy surfaces were observed in simulated synovial environments including phosphate buffer saline (PBS) with pH 7.4, sodium lactate solution with pH 4, and pH 2 adjusted by adding HCl. The cyclic reciprocating motion of an alumina sphere was applied with controlled elastic normal load while monitoring OCP and fretting current during potentiostatic condition. Favorable modifications of oxide chemistry and passive film recovery process were observed on the Ti alloy surface. This result explained why the CoCrMo implant surface showed higher wear rate than that of the Ti6Al4V implant surface.

## 2. Materials Description

CoCrMo (F1537) and Ti6Al4V (F136) rods were manufactured using the electrical discharge machining (EDM) method to rectangular $1.00 \times 2.54 \times 22.95$ mm$^3$ specimens. The EDM method was used to minimize residual stress, and the specimens were embedded in an epoxy disk for electrochemical tests. Small rectangular $1.00 \times 2.54$ mm$^2$ areas on top and bottom of the specimens were exposed out of the epoxy holder, and polished using 600, 1200 and 2400 grit silicon carbide papers, and finally a 0.05 μm colloidal silica and alumina on polishing cloths for 10 to 15 min until a mirror finish was obtained. The microstructure was observed by scanning electron microscopy, and the chemical compositions were analyzed. As illustrated in Figure 1, the CoCrMo surface illustrated large grain sizes in the order of 100 μm. The Ti6Al4V surface consisted of mixed ($\alpha + \beta$) phases. The average concentrations of the major chemical components are summarized in Table 1.

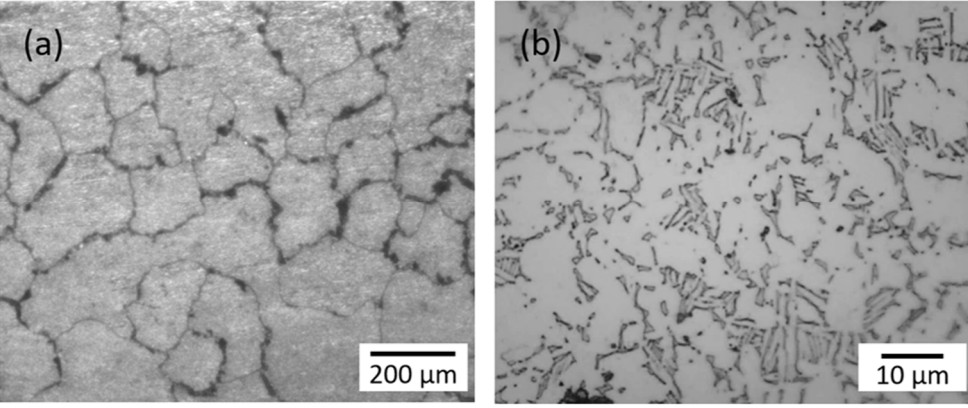

**Figure 1.** Microstructures of (**a**) CoCoMo and (**b**) Ti6Al4V.

**Table 1.** Chemical element concentrations of two metal specimens.

| Ti6Al4V | | Ti | Al | V |
|---|---|---|---|---|
| α phase | wt.% | 89.77 | 6.04 | 4.19 |
| | at.% | 85.96 | 10.26 | 3.78 |
| β phase | wt.% | 86.29 | 5.27 | 8.44 |
| | at.% | 83.29 | 9.04 | 7.67 |
| CoCrMo | | Co | Cr | Mo |
| wt.% | | 27.72 | 26.90 | 06.91 |
| at % | | 52.71 | 25.63 | 3.84 |

Mechanical characterizations were performed before testing, including surface roughness, hardness, and the elastic modulus of both the Ti6Al4V and the CoCrMo samples. The roughness of the surface of the metal alloys was obtained by performing a linear scan using the optical profilometer (Nanovea, Irvine, CA, USA). The surface was scanned at multiple random areas using measuring parameters of 1 mm scan length with 4 μm resolution. The hardness, elastic modulus and yield strengths were obtained using the nanoindenter (Nanovea, Irvine, CA, USA). The samples were indented using a standard Berkovich diamond tip at a peak load of 200 mN and loading rate of 300 mN/min. Characterization results for the respective metal alloys are summarized in Table 2.

**Table 2.** Characterization of implant specimens.

| Materials | Roughness ($R_a$, nm) | Elastic Modulus (GPa) | Hardness (GPa) |
|---|---|---|---|
| Ti6Al4V | $36 \pm 8$ | $134 \pm 21$ | $5.07 \pm 0.25$ |
| CoCrMo | $40 \pm 12$ | $299 \pm 13$ | $6.96 \pm 0.16$ |

*Experimental Details*

A custom tribochemical bath was designed to conduct the electrochemical tests during active sliding contact against a ceramic ball. This chemical bath consisted of the specimen in the epoxy disk, a rubber ring preventing leakage of electrolyte between the epoxy disk and chemical bath, an aluminum foil to ensure electrical contact of the specimen to the Reference 600 potentiostat controller (Gamry Instruments, PA, USA), and a bath component providing stable electrical contacts of three electrodes including the working electrode (the exposed surface of the specimen), the reference electrode, and the counter electrode. The large cylindrical space in the bath component accommodated a sufficient amount of simulated synovial fluid in contact with the specimen surface and two electrodes. The test bath components were manufactured out of a chemically inactive nylon block. A saturated calomel reference electrode (SCE) and a platinum wire (Gamry Instruments, PA, USA) were employed to detect the electrochemical signals from the specimen surfaces. A silver paste was applied to ensure full electrical connection between the specimen and the aluminum foil. Schematics of the test bath are presented in Figure 2a,b. This electrochemical bath was mounted in the nano tribometer (Nanovea, CA, USA) to perform reciprocal sliding contact experiments while monitoring electrochemical responses and coefficients of friction changes (Nanovea, CA, USA).

The simulated synovial fluids used the corrosion test included phosphate buffered saline (PBS) solution and sodium lactate solutions. The PBS solution consisted of NaCl 9 g/L, $KH_2PO_4$ 0.144 g/L, and $Na_2HPO_4$-$7H_2O$ 0.795 g/L. The sodium lactate (SL) solution was made by 0.1 mole of sodium lactate ($C_3H_5NaO_3$) powder dissolved in 250 mL deionized water with diluted 0.1 mole of hydrochloric acid to adjust the acidity levels to pH 4 and pH 2. The PBS and SL solutions represented healthy synovial and inflammatory synovial environments, respectively [9–11]. A 3-mm diameter alumina sphere was used to apply sliding contact loads on the exposed surface of the specimen. In order to determine the elastic range normal contact force for the sliding contact fatigue experiment, Hertzian contact stress was calculated using Young's modulus (E = 310 GPa) and Poisson's ratio (ν = 0.21) of Alumina balls. The elastic normal load of 143 mN applies contact pressures of approximately 45% of the yield strength of CoCrMo and 40% of the yield strength of Ti6Al4V. The sliding contact fatigue test was carried out for a time period of 60 min that corresponded to 1800 cycles of reciprocations. The wear test parameters are summarized in Table 3.

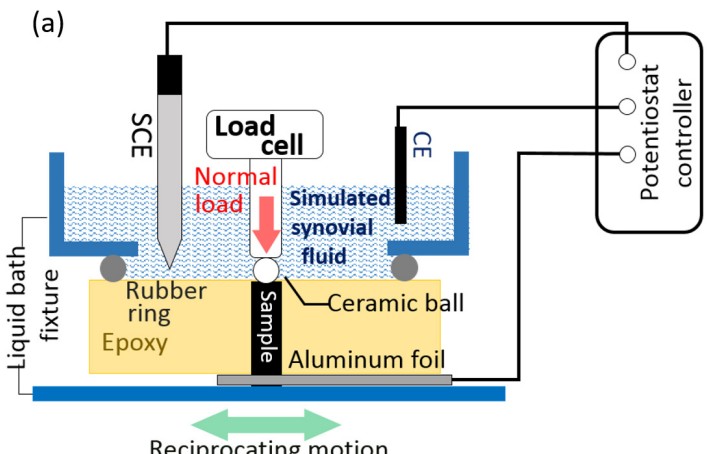

**Figure 2.** (**a**) Schematic of a tribocorrosion experimental set up that consists of a nanoindenter-based wear tester and a synovial fluid bath; (**b**) Specimen prepared in epoxy and liquid cell components.

**Table 3.** Corrosion fatigue test parameters.

| Method | Parameters |
| --- | --- |
| Contact mode | Reciprocating |
| Contact load | Constant normal at 143 mN |
| Sliding distance | 200 μm |
| Sliding speed | 12 mm/min |
| Sliding cycles | 1800 cycles |
| Environment | PBS (pH 7.4), SL (pH 4 and 2) |

## 3. Results and Discussion

Electrochemical tests included open circuit potential and current density with applied static potential during active sliding contact on both the CoCrMo and the Ti6Al4V surfaces. The experiments were repeated in different simulated synovial environments. The PBS solution was considered as a normal healthy body fluid, and sodium lactate solutions with two different pH levels were considered as unhealthy inflammatory environments.

### 3.1. Open Circuit Potential Measurement

The damaged passive film by sliding contact resulted in a significant potential drop in the negative, followed by a subsequent increase in anodic current [36]. The coefficients of friction were monitored with the measured OCP changes. The observations illustrated the effect of the synovial environments on the mechanical wear and corrosion damage. Therefore, evolution of the potential and current during continuous sliding explained the metal oxidation chemistry of the alloys against wear and corrosion.

### 3.1.1. Phosphate Buffer Saline (PBS) Solution pH 7.4

The potential change by reciprocating motion, with a constant normal load of 143 mN, was monitored before sliding (dwell), during sliding (active articulations) and after sliding motion (recovery) on the CoCrMo and the Ti6A14V. As shown in Figure 3, the 1-h dwell time was allowed to stabilize, and then the reciprocating motion using the spherical alumina was applied at 143 mN normal force. There was a significant potential drop on both the CoCrMo and the Ti6Al4V surfaces due to the exposure of pure metal, via a damaged oxide layer, to the PBS solution. The potential of the Ti6Al4V dropped instantaneously and gradually increased (positively changed) during continuous reciprocations. The potential of the CoCrMo progressively decreased to negative, and continued at −0.4 V until the active contact ceased. It was evident that the potential of the Ti6Al4V dropped as soon as the slider motion was initiated, but spontaneous repassivation occurred even during active sliding. The potential gradually recovered to the level of its initial dwell potential as the sliding contact continued. It was notable that the presence of the PBS was beneficial to the Ti6Al4V to reform its stable oxide layer. However, during the fatigue wear on the CoCrMo, the potential continuously dropped until the sliding motion ceased. This implied that the chromium oxide layer was progressively degraded by continuous abrasion by the slider head. The chemomechanical response illustrated that the reformed chromium oxide layer was not as stable as the original passive layer formed in ambient. During the sliding fatigue on the CoCrMo surface, rapid fluctuations of OCP response illustrated a repetitive process of breakage-reformation cycles of the oxide layer in the PBS [27–29]. However, after a dramatic potential drop upon initiating contact stroke, the Ti6Al4V surface presented an increase in potential (positive change). This result on the Ti6Al4V surface implied spontaneous titanium oxide recovery during contact corrosion fatigue. After the sliding contact stopped, the CoCrMo surface was repassivated quickly, while the Ti6Al4V was readily repassivated during sliding motions, and continued the recovery process.

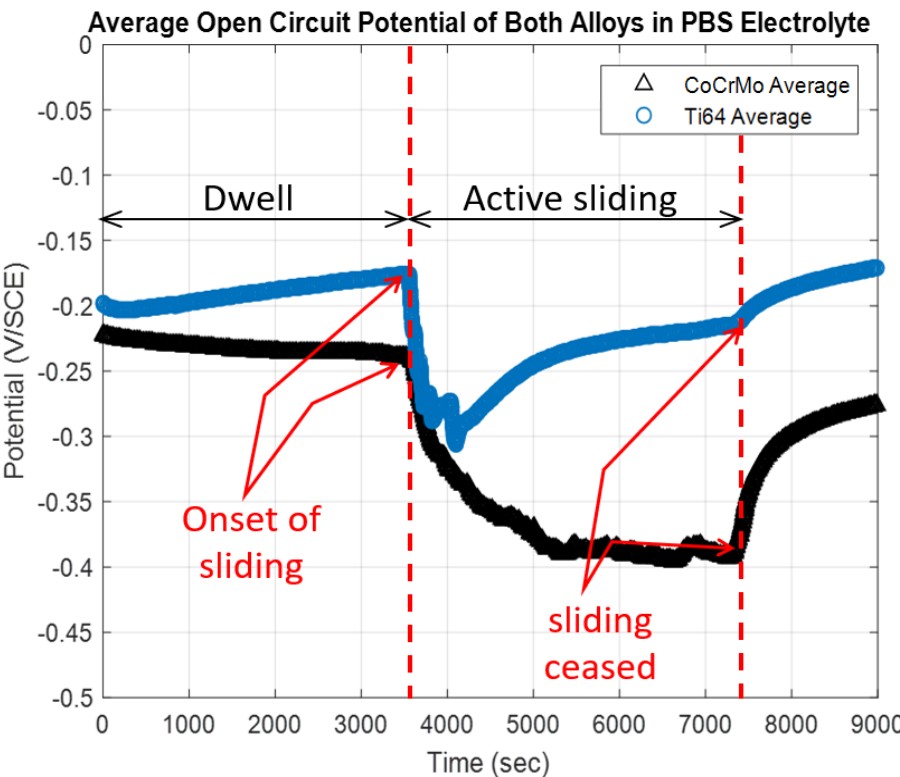

**Figure 3.** Open circuit potential response during the corrosion fatigue test on CoCrMo and Ti6Al4V in phosphate buffered saline (PBS) solution.

Figure 4 summarizes the OCP responses. Negative potential drops were found on the CoCrMo surfaces at the onset of sliding as well as during the active sliding contacts. However, it is evident that the positive change of OCP on the Ti6Al4V illustrated prompt reformation of chemically protective titanium oxide during the active sliding contacts.

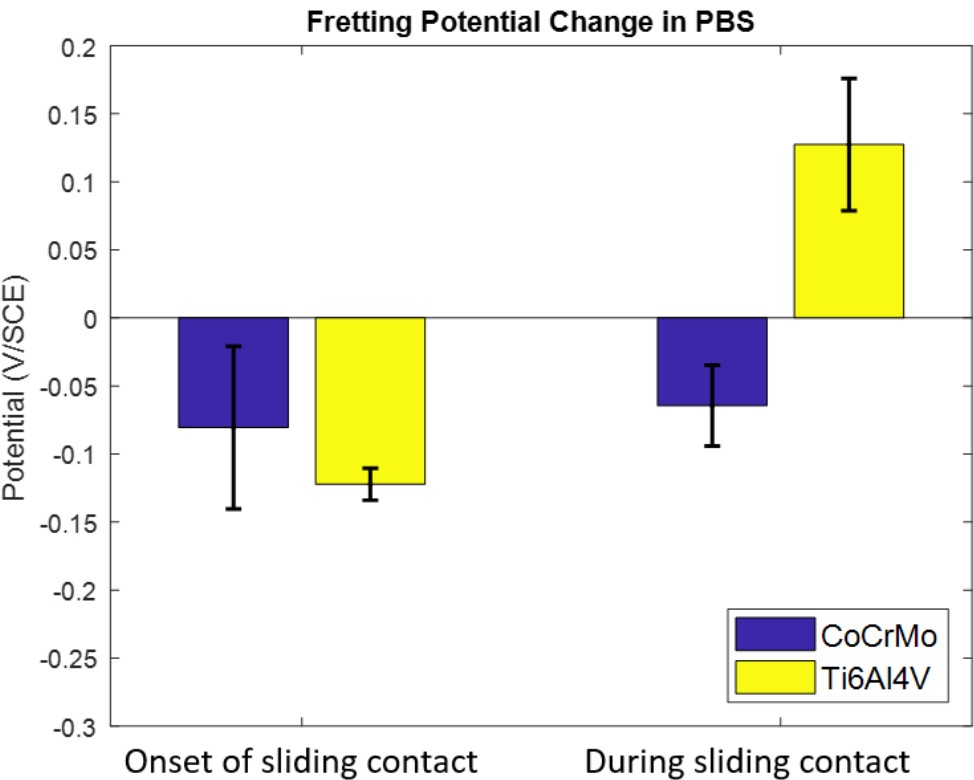

**Figure 4.** The effect of the PBS environment on OCP response on the CoCrMo and the Ti6Al4V during reciprocating sliding contact.

Coefficients of friction (CoF) values were monitored during the sliding test. The positive and negative values of CoF in Figure 5 dictate forward and backward sliding motions, respectively. The results present greater average CoF values on the CoCrMo surface. Throughout the course of sliding contact, CoF variations on the CoCrMo showed unstable frictional interaction. This result was in accordance with the significant oxide damage from OCP responses on the CoCrMo in Figure 3. The CoF values on the Ti6Al4V rapidly evolved in the early stage of reciprocating in the range of 0.4 to 0.6, but soon stabilized to a value less than 0.6 in the cumulative sliding distance of 100 mm that corresponded to 250 reciprocated cycles.

The CoF values were explained by micro-optical image inspections of wear damage. As shown in Figure 6, the wear debris continuously produced from the thin chromium oxide layer by each sliding stroke continually affected the friction forces on the CoCrMo surface. However, on the Ti6Al4V surface, the sliding motion led to successive plastic deformation on the top layer and the subsurface crack under the deformed layer would be propagated during sliding cycles. Larger agglomerated wear particles are indicative of more ductile behavior on Ti6Al4V surfaces. Therefore, changes in OCP and CoF responses on the Ti6Al4V presented less sensitivity during microcracking under the oxide layer.

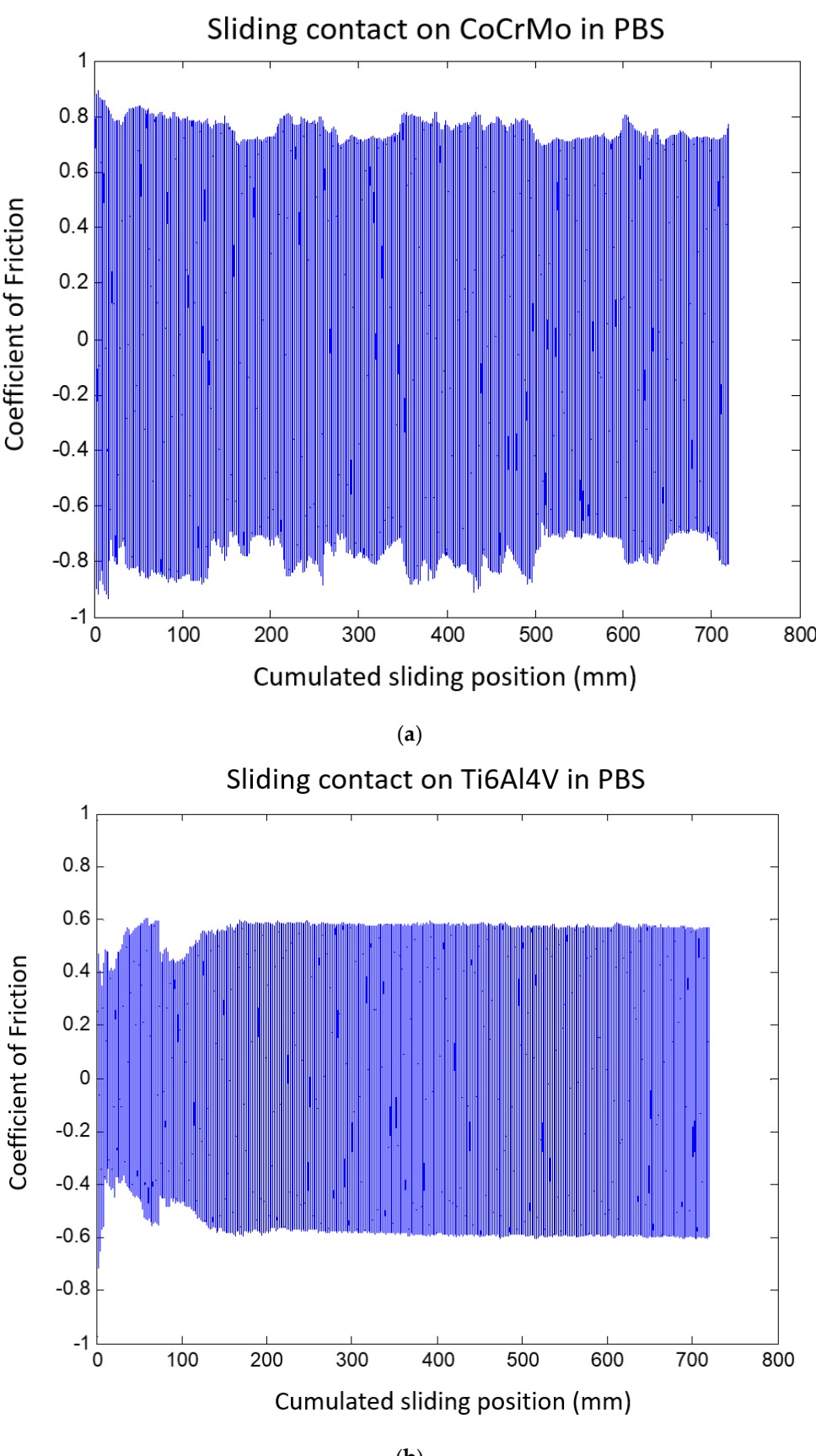

(**a**)

(**b**)

**Figure 5.** (**a**) Coefficient of friction of the CoCrMo during reciprocating contact in the PBS solution. (**b**) Coefficient of friction of the Ti6Al4V during reciprocating contact in the PBS solution.

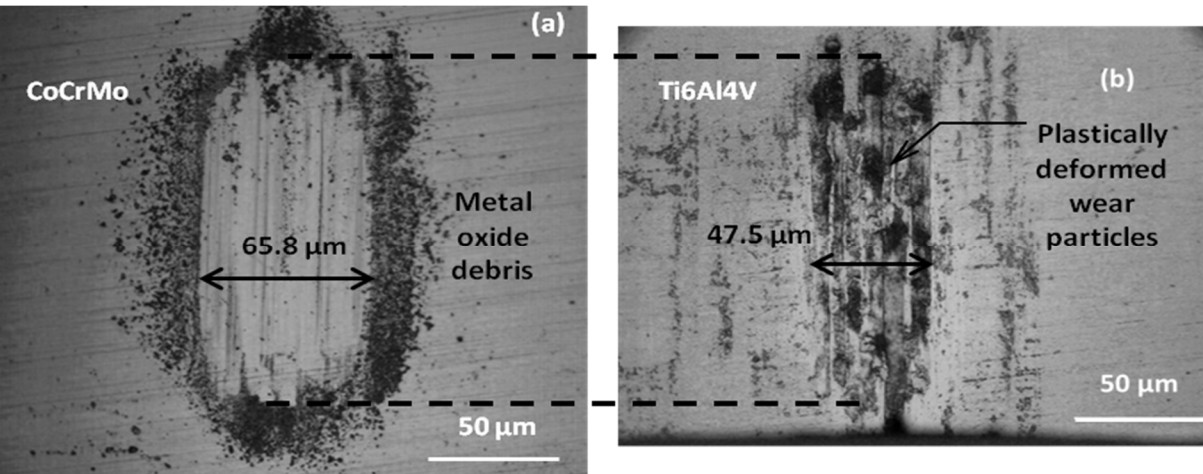

**Figure 6.** Representative micro images of wear tracks on (**a**) CoCrMo surface producing nano size oxide debris and (**b**) Ti6Al4V surface forming wear particle agglomerations by reciprocal sliding contact in PBS.

3.1.2. Sodium Lactate pH 4

The potential change by reciprocating motion with a constant normal load was monitored during sliding contact fatigue on the CoCrMo and the Ti6A14V in a sodium lactate solution. The pH level of the lactate solution was adjusted by adding hydrochloric acid to obtain a controlled acidic environment of pH 4.0. As shown in Figure 7, the overall fatigue corrosion processes in sodium lactate pH 4 presented similar behaviors in PBS pH 7.4. However, there was a significant potential drop on the CoCrMo and a reduced potential drop on the Ti6Al4V surfaces due to the exposure of metal via a damaged oxide layer to the sodium lactate pH 4 solution. During continuous sliding contact on the CoCrMo, the potential continuously dropped until the sliding motion ceased. The potential drop on the CoCrMo surface accelerated up to 5000 s (approximately 1000 cycles), and later decelerated in the rest of the reciprocations, whereas the potential of the Ti6Al4V surface instantaneously decreased and maintained the potential values throughout the cyclic sliding. It was evident that the potential of the Ti6Al4V slightly dropped as soon as the slider motion was initiated, but that it was followed by very steady repassivation. The direct comparison of magnitudes of the potential drop between the CoCrMo and the Ti6Al4V clearly described the different oxidation chemistry processes. It was notable that the presence of the moderate acidic lactate improved the corrosion protection of the Ti6Al4V surface. This result implies that the tribological stability and rapid regrowth of Ti6Al4V is superior to those of CoCrMo in the moderate acidity of lactate fluid. In particular, it is presumable that the fatigue corrosion on the CoCrMo surface was accelerated in the sodium lactate pH 4. This could be explained by the rate of repassivation of titanium oxide being faster than the rate of metal ion dissolutions in the presence of lactate medium. After the sliding cycles stopped, the CoCrMo surface was rapidly repassivated, while the Ti6Al4V was readily under the progressive repassivation during contact fatigue. Therefore, the Ti6Al4V presented desirable oxide chemistry that recovered the damaged surface in the sodium lactate pH 4 solution.

Contact corrosion fatigue responses in a sodium lactate pH 4 environment are summarized in Figure 8. The drastic potential drop on the CoCrMo was observed when sliding contact was initiated and continually lowered during cyclic motions. Potential changes on the Ti6Al4V in sodium lactate were similar to those in PBS, but the positive potential change during contact cycles was reduced when compared to the result from corrosion fatigue in PBS.

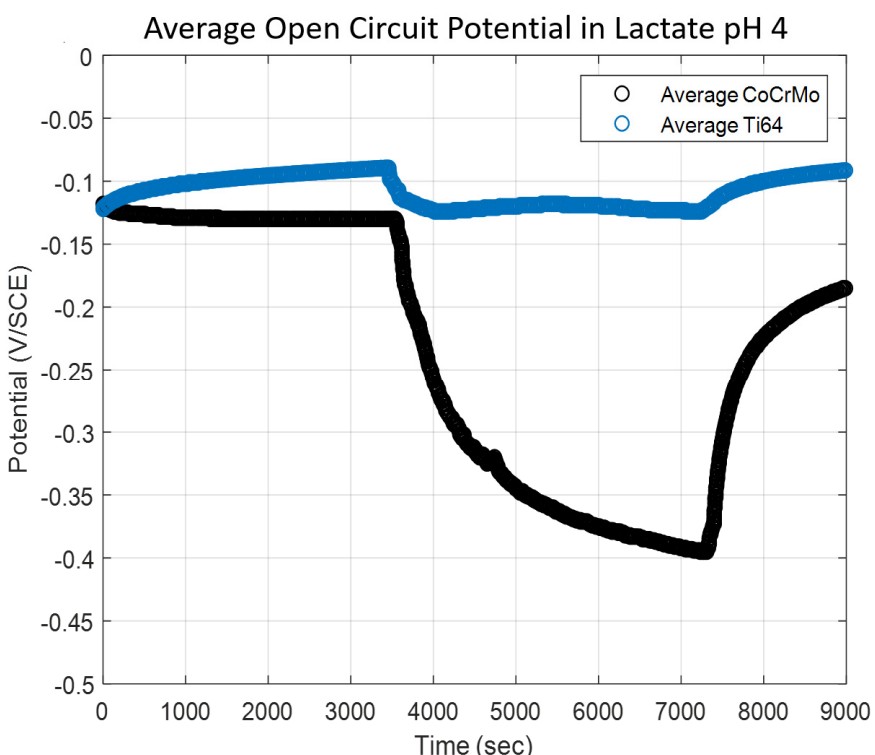

**Figure 7.** Open circuit potential change by reciprocal contact in sodium lactate pH 4.

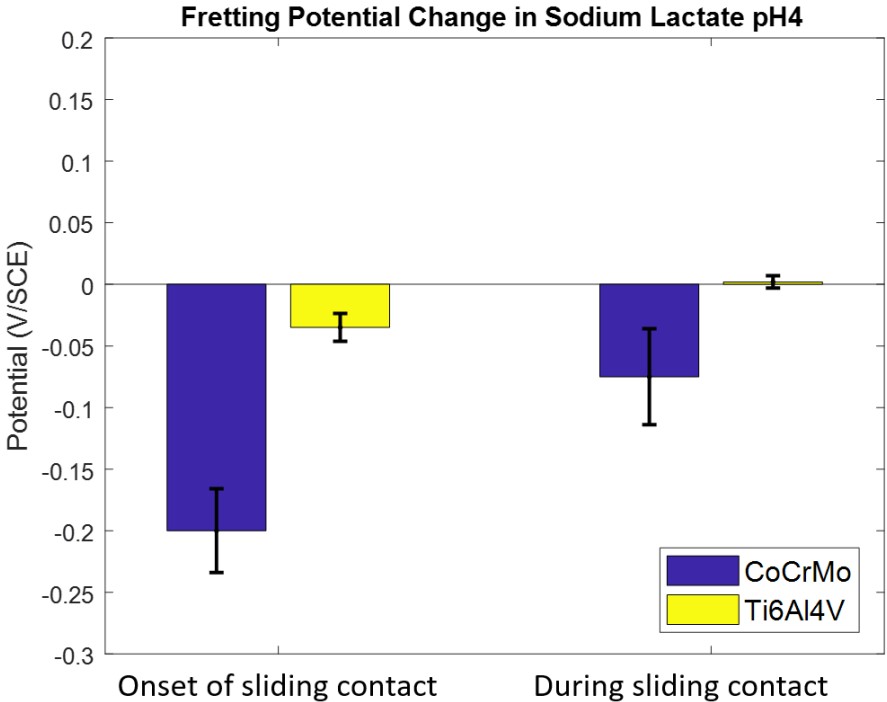

**Figure 8.** The effect of a sodium lactate pH 4 environment on OCP response on CoCrMo and Ti6Al4V during reciprocating sliding contact.

Friction responses were significantly modified in the sodium lactate environment as presented in Figure 9a,b. Average CoF values on both surfaces were reduced in the sodium lactate pH 4 compared to the results from tests in the PBS. However, similarly to PBS, the average CoF values on the Ti6Al4V surface presented less than the average CoF values on the CoCrMo in sodium lactate pH 4. It was noticed that the CoF values on the CoCrMo

reached a steady level at approximately 0.5 after 30 mm cumulated sliding distance. It is evident that the CoF change is associated with simultaneous modifications in OCP drops. During the steady state kinetic friction, OCP continually decreased on CoCrMo surface. However, Ti6Al4V surface showed stable responses on both friction coefficients and OCP signals during sliding contact.

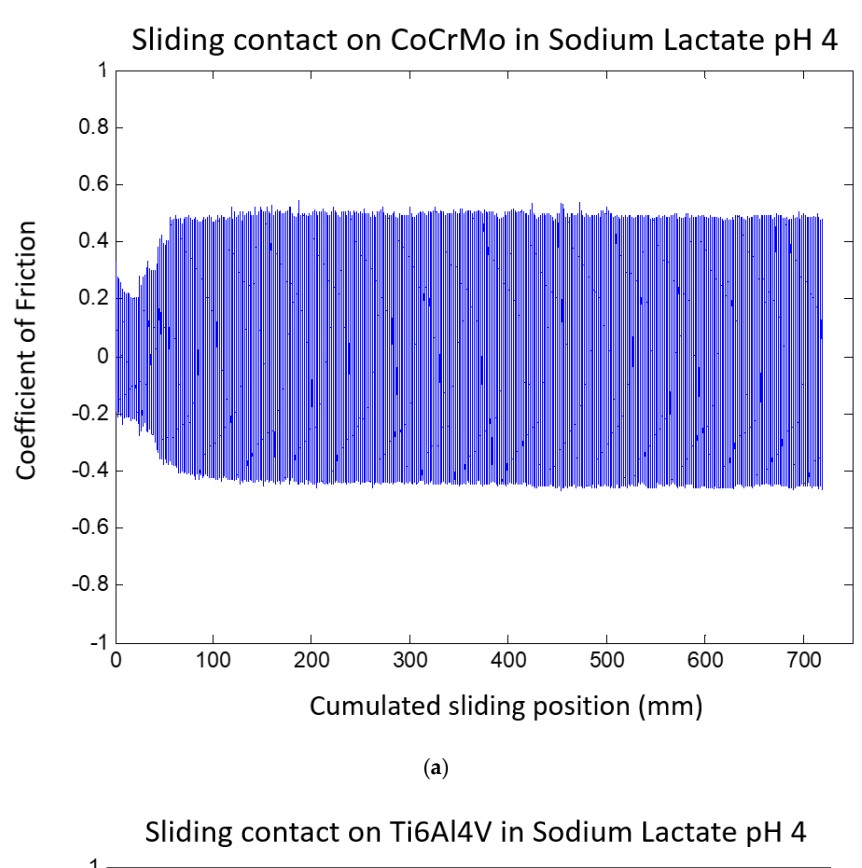

(**a**)

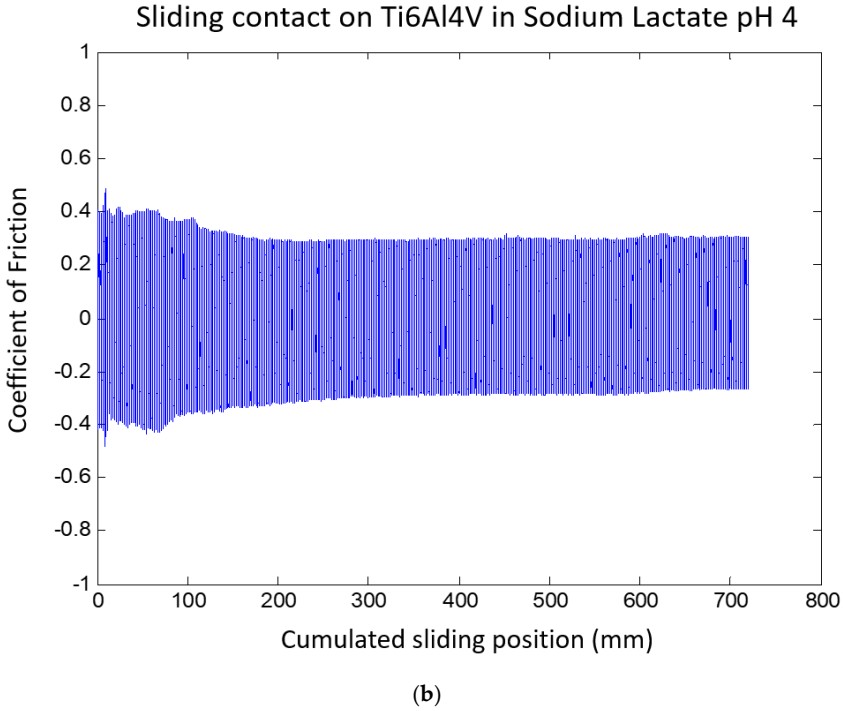

(**b**)

**Figure 9.** (**a**) Coefficient of friction of CoCrMo during reciprocal sliding contact in sodium lactate pH 4. (**b**) Coefficient of friction of Ti6Al4V during reciprocal sliding in sodium lactate pH 4.

### 3.1.3. Sodium Lactate pH 2

The potential change by reciprocating motion with a constant normal load of 143 mN was monitored in the sodium lactate pH 2. As shown in Figure 10, the potential was allowed to stabilize for an hour, and then the reciprocating motion using the spherical alumina was applied. The potential change of the Ti6Al4V surface in the sodium lactate pH 2 illustrated a dramatic drop as soon as the sliding was initiated, and then during the continuous contact the potential was increased (positive change) in the rest of the test period. In this chemically aggressive environment, potential response varied, and repassivation did not effectively occur on the Ti6Al4V surface. In contrast, the CoCrMo surface presented a small potential drop when the mechanical contact started, but the potential proportionally decreased (negative change) until the sliding was ceased. However, there was a significant change in potential drops in this reactive lactate pH 2 solution. More importantly, the maximum potential drop on the Ti6Al4V during sliding contact was greater than that on the CoCrMo surface. In the other corrosion fatigue tests in PBS and sodium lactate pH 2, the potential drop was more significant on the CoCrMo surface than on the Ti6Al4V surface. This result described how the corrosion sensitivity of the Ti6Al4V was anomalously manifested in the reactive acidic environment. However, the corrosion process was clearly accelerated on the CoCrMo surface, while the recovery process of oxidation on the Ti6Al4V was observed during the course of the sliding contacts. Figure 11 illustrates that the rate of oxide recovery on Ti6Al4V in a reactive environment is not as much as in the neutral PBS pH 7.4. The initial potential drop on the CoCrMo was reduced in an aggressive acidic condition, but the continuous potential drop during the fretting on the CoCrMo was greater in lactate pH 2 when compared to the potential changes in PBS and lactate pH 4 solutions. The potential kinetics of the Ti6Al4V illustrated that if greater potential drop at the onset of fretting was obtained, rapid recovery oxidation took place during contact corrosion fatigue.

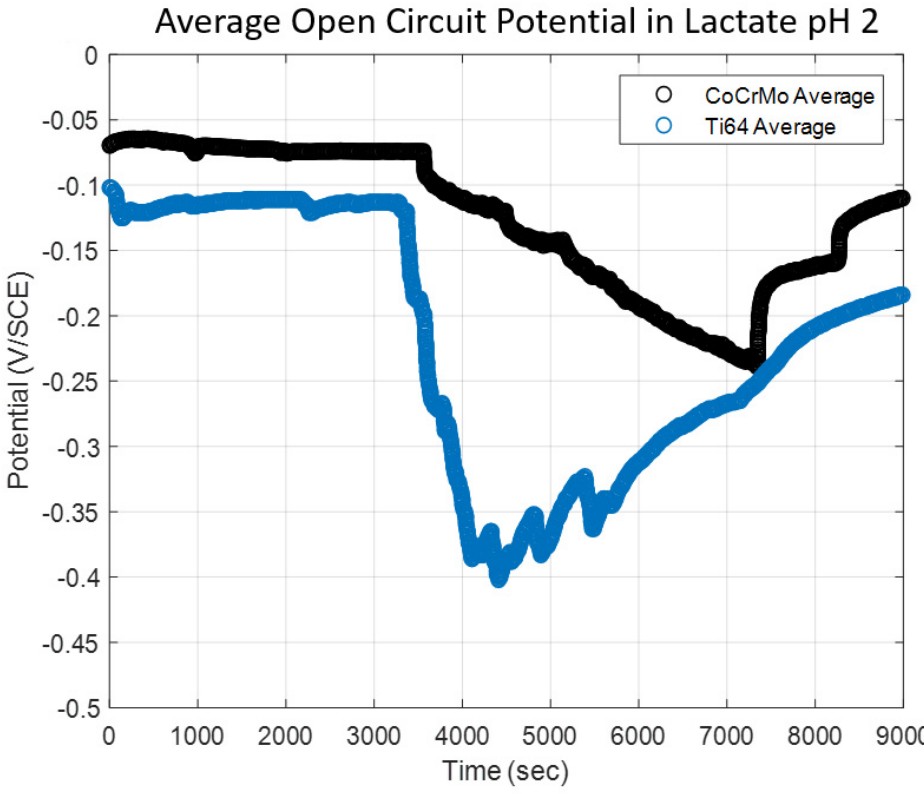

**Figure 10.** The effect of a sodium lactate pH 2 environment on OCP response on CoCrMo and Ti6Al4V during reciprocating sliding contact.

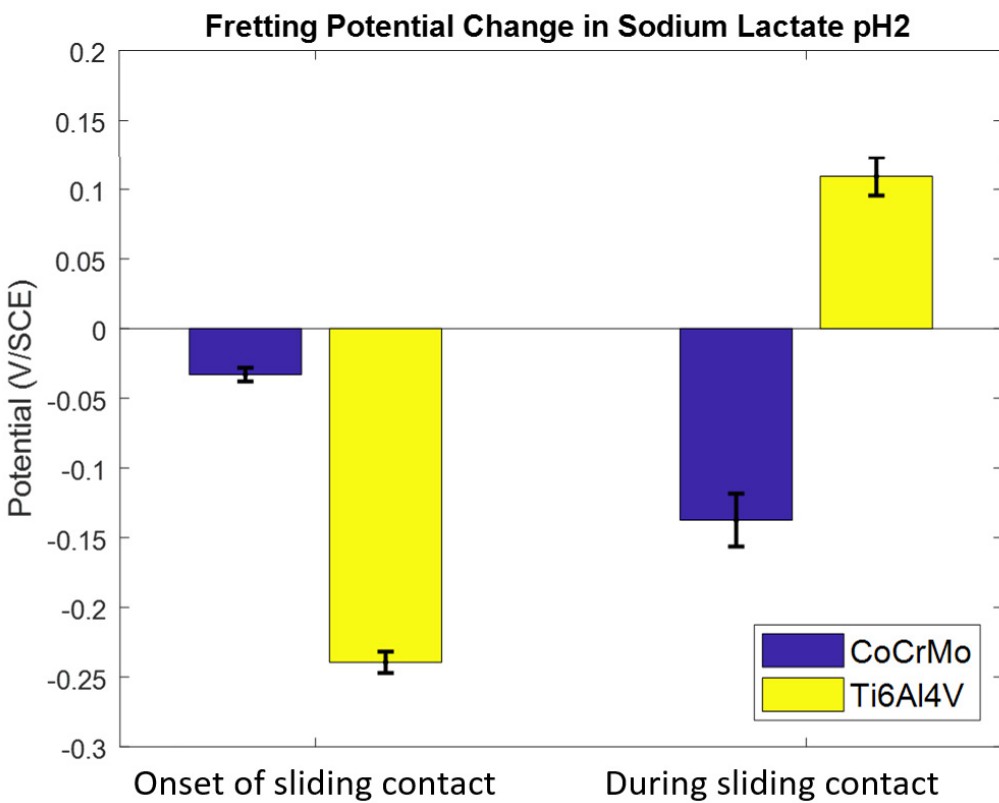

**Figure 11.** The effect of a sodium lactate pH 2 environment on OCP response on CoCrMo and Ti6Al4V during reciprocating sliding contact.

The CoF values on the CoCrMo progressively increased during the course of sliding cycles, as presented in Figure 12. The gradual increase in the COF values on the CoCrMo may describe the continuous potential drop during fatigue contact. The average COF values of the Ti6Al4V were greater than those of the CoCrMo in this environment. This result was in accordance with the OCP results; there was a greater potential drop on the Ti6Al4V than on the CoCrMo. At the maximum potential drop on the Ti6Al4V surface, high potential variations were observed. This result was well illustrated in the CoF curves in Figure 12. Interestingly, the average COF values were slightly reduced on the Ti6Al4V surface, while the CoF values on the CoCrMo surface were significantly reduced in lactate pH 2. Figure 13 summarizes the average COF values in three different environments. It presents less sensitivity of CoF values on the Ti6Al4V surface with the different synovial environments. In the reactive solutions of lactate pH 4 and pH 2, the average COF was almost the same. The COF values of the CoCrMo revealed the significance of the environments. With increased acidity, average COF values were reduced.

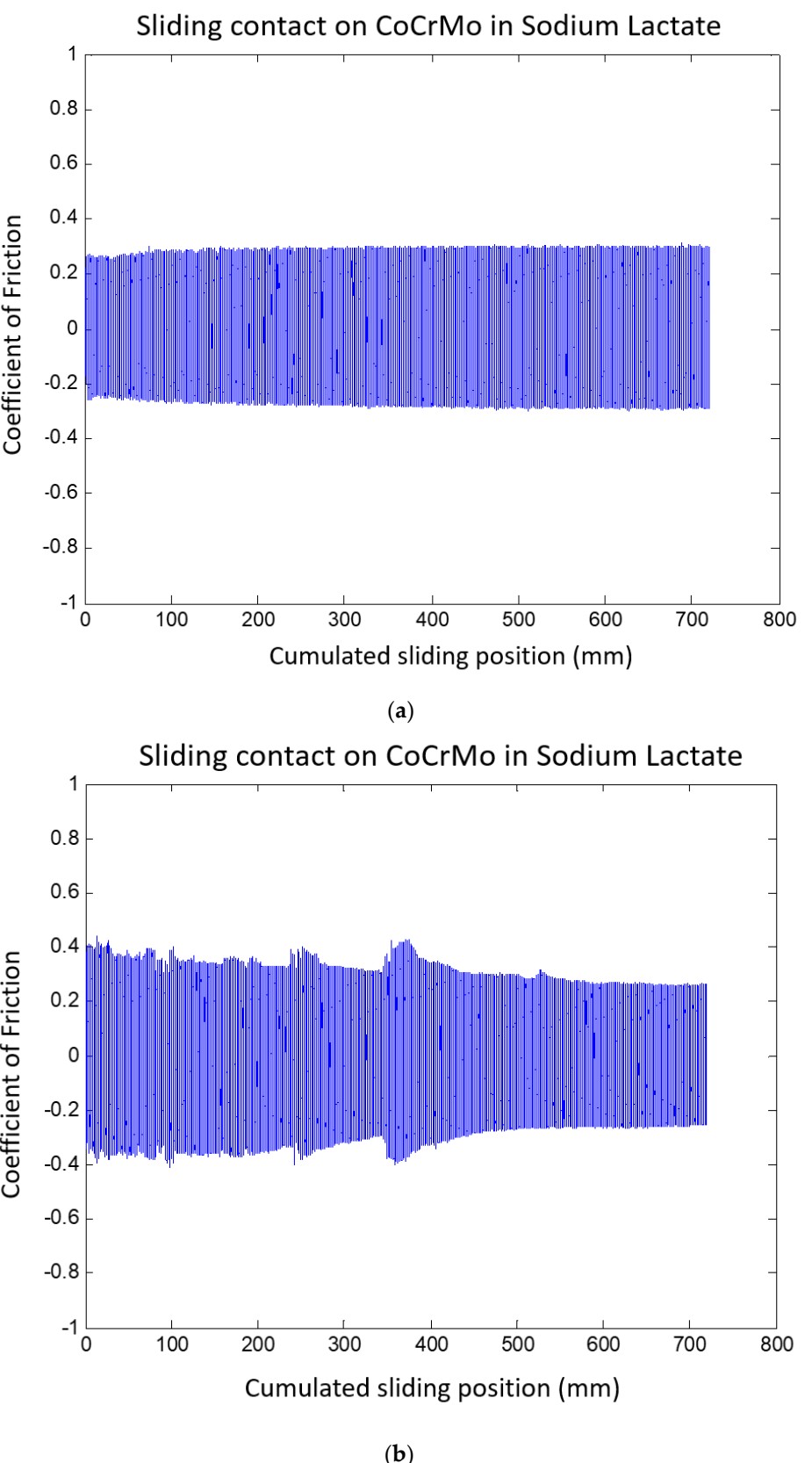

**Figure 12.** (**a**) Coefficient of friction of CoCrMo during reciprocal sliding in sodium lactate pH 2. (**b**) Coefficient of friction of Ti6Al4V during reciprocal sliding in sodium lactate pH 2.

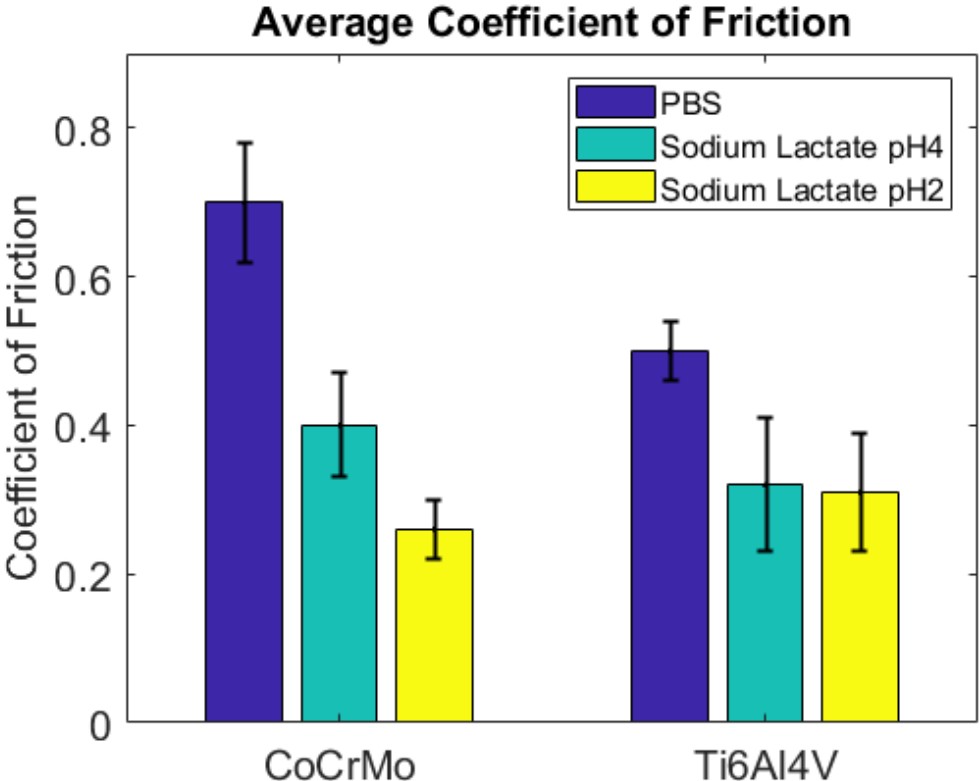

**Figure 13.** Average coefficient of friction in different simulated synovial environments.

*3.2. Fretting Static Current*

Potentiostatic polarization tests were performed at a series of potential voltages at
−0.6, −0.45, 0.0, +0.45 V and +0.6 V in all three environments on both alloys during
reciprocal sliding contact. Before sliding contact started, current responses were allowed
to stabilize to obtain a baseline current up to 900 s. The baseline current was measured
to quantify the change in current when the reciprocal motion was initiated. Figure 14
is a representation of fretting current change against potentials −0.6, 0.0 and +0.6 V in
PBS. On both surfaces with all negative and positive potentials, there was no significant
current change in PBS during fatigue contact. Small positive peaks on current curve in
positive potential (+0.6 V) were observed only on the CoCrMo surface. Current responses
of the Ti6Al4V were smooth and continuous throughout the tests. Therefore, the results
concluded that there would not be aggressive fretting corrosion damage in PBS. However, in
sodium lactate pH 4 shown in Figure 15, significant current changes on the CoCrMo surface
were found when zero or positive potentials were applied. Similarly, Ti6Al4V surfaces
showed smooth and continuous fretting current in sodium lactate pH 4,. There was no
distinctive current change on Ti6Al4V induced by sliding fatigue contact. This implies
that the CoCrMo surface was more actively corroded during fictional sliding without and
with positive potentials in an acidic environment. In sodium lactate pH 2 as illustrated in
Figure 16, amplified current signals were detected on the CoCrMo surface with zero and
positive potentials. Similarly, in PBS and sodium lactate pH 2 solutions, Ti6Al4V surfaces
showed insignificant current changes induced by sliding contact. These potentiostatic test
results agreed with the OPC test results during contact in simulated synovial environments.
Hence, this may imply that the CoCrMo surface was vulnerable to fretting corrosion in
reactive acidic conditions, while the Ti6Al4V would be more stable against the combined
mechanical and electrochemical stimuli.

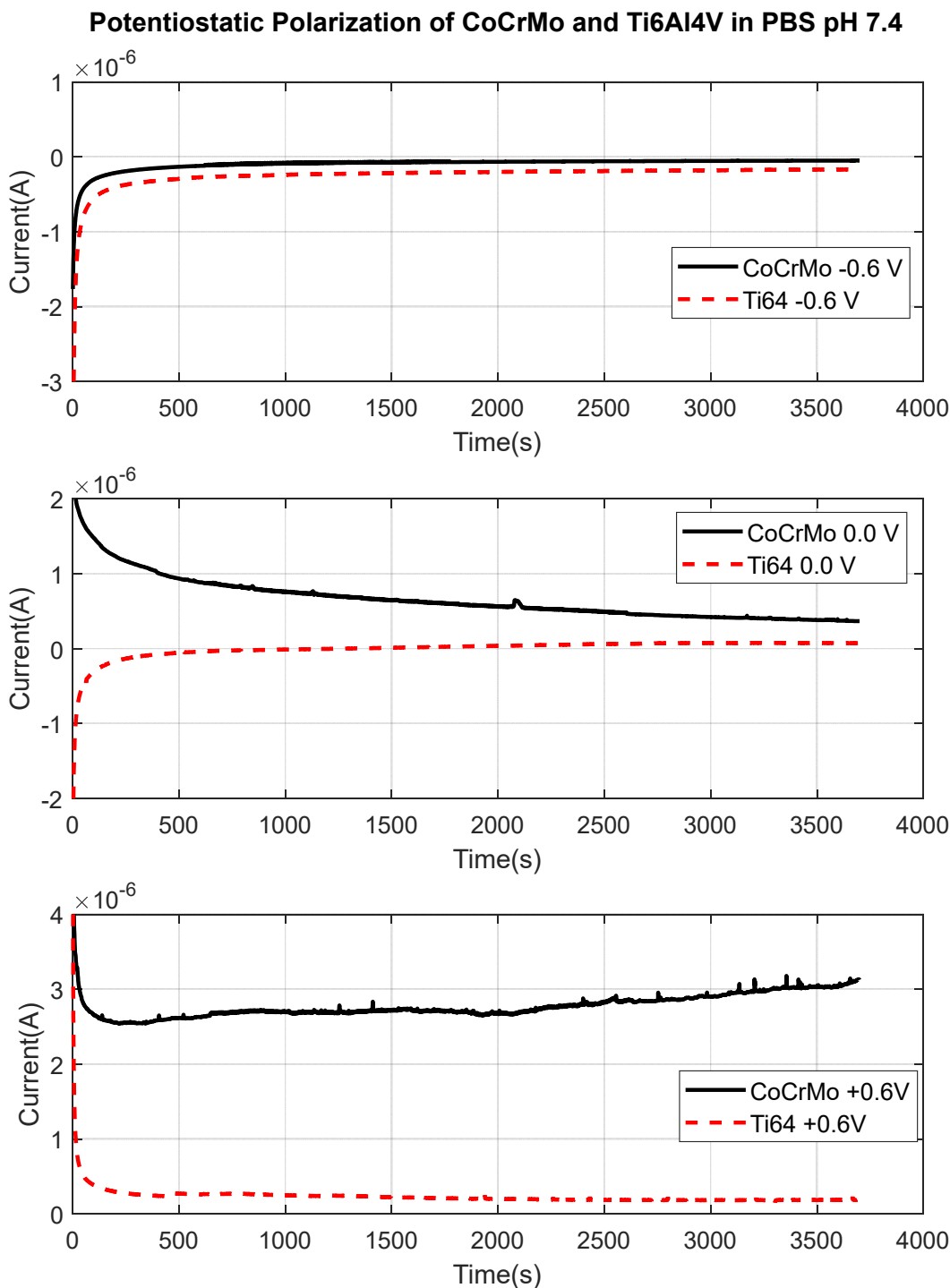

**Figure 14.** Fretting current changes through the disturbed areas by reciprocal sliding in PBS with applied potentials. Active surface areas of CoCrMo and Ti6Al4V are $3.086 \times 10^4$ µm$^2$ and $3.046 \times 10^4$ µm$^2$, respectively after 1800 sliding cycles.

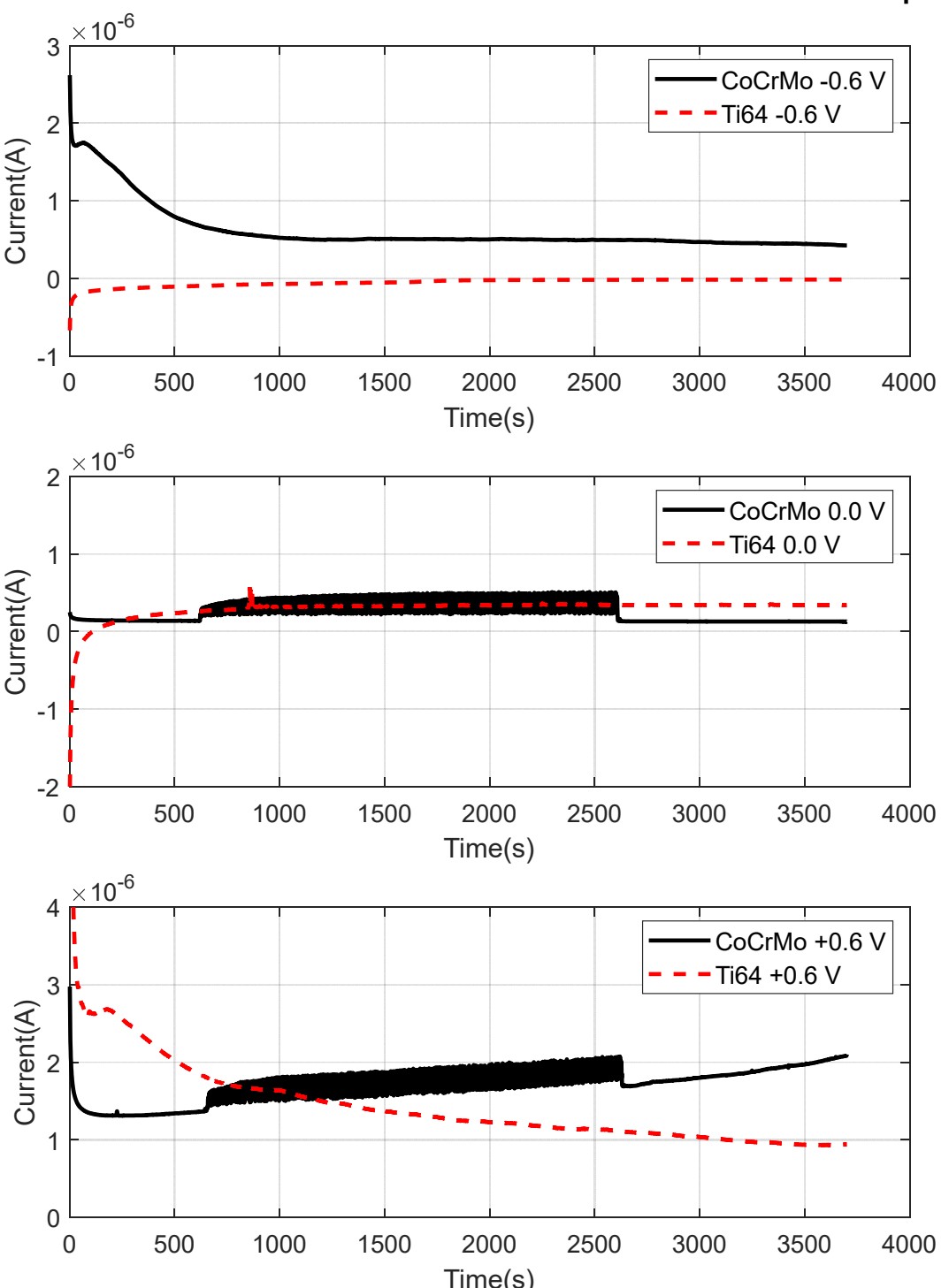

**Figure 15.** Fretting current changes through the disturbed areas by reciprocal sliding in in lactate pH 4 with applied potentials. Active surface areas of CoCrMo and Ti6Al4V are $3.820 \times 10^4$ μm$^2$ and $3.410 \times 10^4$ μm$^2$, respectively after 1800 sliding cycles.

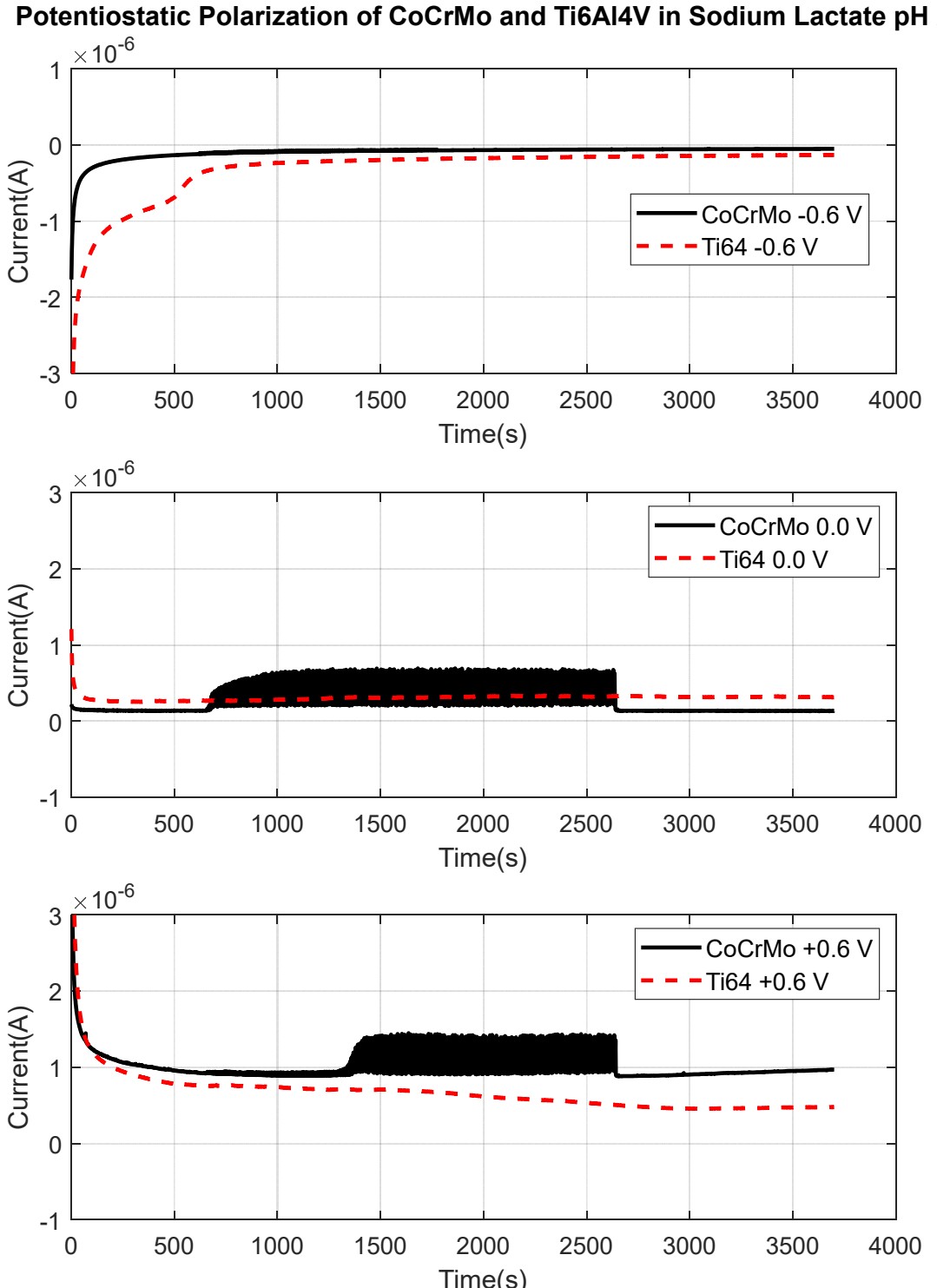

**Figure 16.** Fretting current changes through the disturbed areas by reciprocal sliding in in lactate pH 2 with applied potentials. Active surface areas of CoCrMo and Ti6Al4V are $5.930 \times 10^4$ $\mu m^2$ and $4.085 \times 10^4$ $\mu m^2$, respectively after 1800 sliding cycles.

The current signals were normalized by the contact damaged area. The wear damage areas were evaluated from the micro images taken after the sliding tests. The fretting current density is summarized in Figure 17a,b. It was evident that positive potential increased fretting corrosion damage, while negative potential suppressed fretting corrosion damage on the CoCrMo surfaces. This result illustrated a strong linear proportionality of

fretting current density with potentials. In particular, the CoCrMo's fretting current density drastically increased in the aggressive acidic lactate solution.

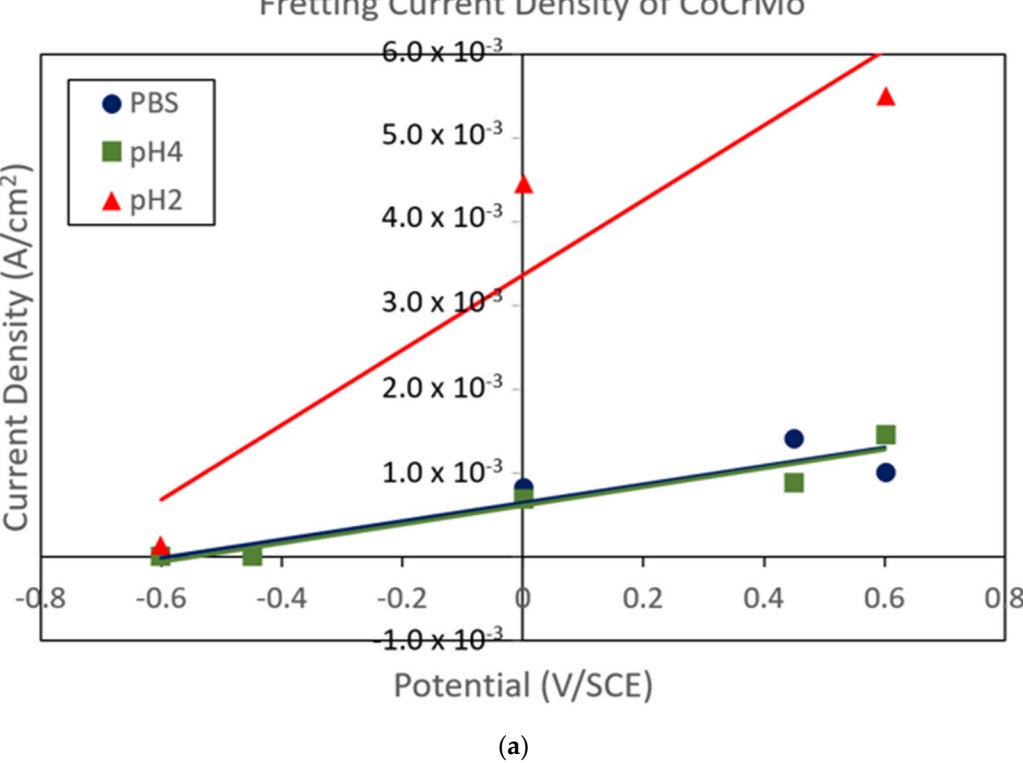

(**a**)

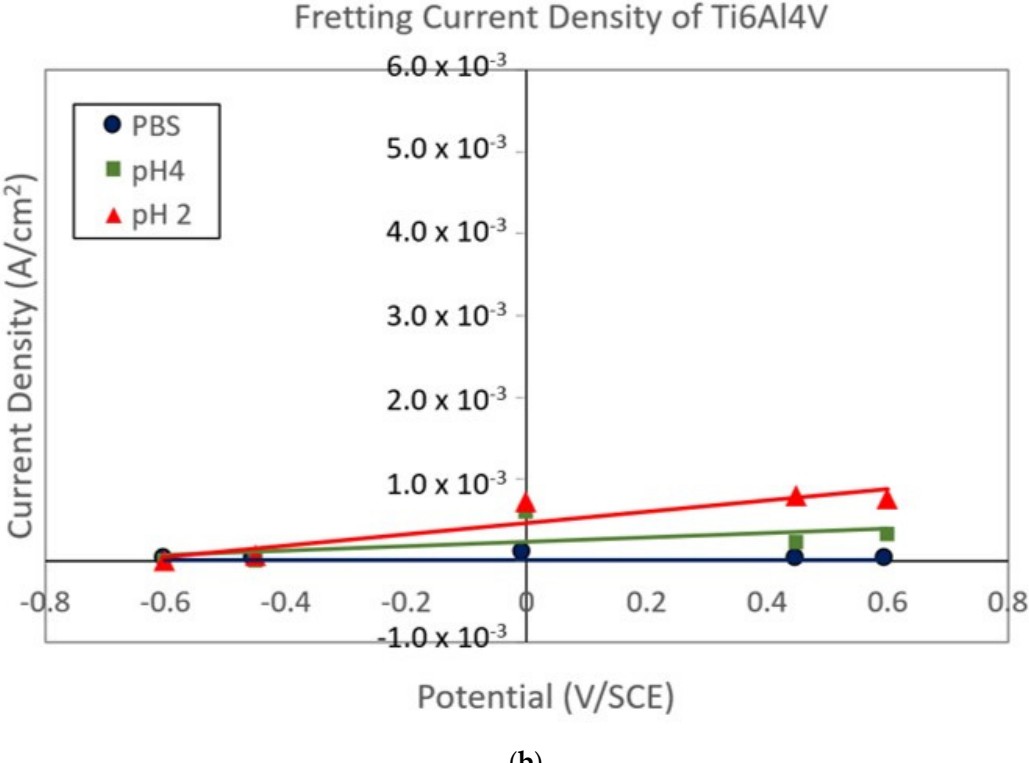

(**b**)

**Figure 17.** (**a**) Fretting current density on the CoCrMo surface with variable potentials applied. (**b**) Fretting current density on the Ti6Al4V surface with variable potentials applied.

Fretting corrosion damage was not significant on the Ti6Al4V surfaces with all potentials. The overall fretting current density values of the Ti6Al4V were only in the range of 15~17% from fretting current density of the CoCrMo. The result illustrated a week linear proportionality of fretting current density with potentials on Ti6Al4V. Similarly, the fretting current density increased with acidity of the solutions. The results prove that the oxidation chemistry of titanium is not sensitive against electrochemical and mechanical stimuli. We conclude that Ti6Al4V implants would be more stable against tribocorrosion damage process.

## 4. Conclusions

This study investigated two commonly used metallic hip implant materials in simulated synovial environments to compare the effects of variable electrochemical environments on the wear mechanism of two different metal alloys. The results of the experiment illustrated that during reciprocating sliding contact, there was a gradual drop in OCP of the CoCrMo alloy, which signified depletion of the chromium oxide layer until fretting ceased. However, after the sliding contact stopped, the protective metal oxide layer was quickly reformed. The results on the Ti6Al4V, showed that there was a significant potential drop at the onset of frictional sliding but that immediate recovery of its oxide layer took place even during the continuous rubbing motions. Therefore, it illustrated that the mechanical strength of the CoCrMo was greater than Ti6Al4V, while the contact fatigue strength of Ti6Al4V was superior. The electrochemical stability of the chromium oxide layer was gradually degraded during the course of reciprocations in all the solutions, while spontaneous regrowth of the titanium oxide layer was evident against mechanical and electrochemical stimuli. This wear process showed that the sliding contact produced nanoscale abrasion of the chromium oxide layer. The repassivation rate on the CoCrMo surface may not be immediate enough to recover the damaged oxide layer. However, from the experimental results of the morphology of large wear particles, the rapidly recovered titanium oxide from OCP and the potentiostatic tests and smaller COF, we may conclude that the fundamental wear mechanism of Ti6Al4V is the delamination wear process. Continuous shear loadings by sliding contact may lead to microcracking at the subsurface with minor damage on the titanium oxide. Mechanical energy by fretting contact was consumed by oxide layer abrasion on the CoCrMo, while the greater plastic strain on Ti6Al4V accelerated repassivation to protect the surface against subsequent electrochemical dissolutions. Therefore, the experimental observation concluded that Ti6Al4V is superior in terms of tribocorrosion compared to the CoCrMo alloy, even though CoCrMo has superior mechanical strengths. Therefore, Ti6Al4V would be a better material for the purpose of load-bearing orthopedic implant components.

**Author Contributions:** Data curation, E.C., M.V.P. and M.C.; Formal Analysis, E.C., M.V.P. and M.C.; Methodology, E.C., M.V.P. and J.J.R.; Supervision, J.J.R.; Writing original draft, J.J.R.; Writing review & editing, J.J.R. All authors have read and agreed to the published version of the manuscript.

**Funding:** The research was funded by the Youngstown State University Research Council grant and Center for Excellence program.

**Institutional Review Board Statement:** Not applicable.

**Informed Consent Statement:** Not applicable.

**Data Availability Statement:** Not applicable.

**Acknowledgments:** The Authors gratefully acknowledge the financial support of the University Research Council grant and Center for Excellence program at Youngstown State University.

**Conflicts of Interest:** The authors declare no conflict of interest.

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
