# Peer review of "Sliding Corrosion Fatigue of Metallic Joint Implants: A Comparative Study of CoCrMo and Ti6Al4V in Simulated Synovial Environments"

_lubricants, doi:10.3390/lubricants10040065_

Round 1

Reviewer 1 Report

The work presents investigations on two types of metallic hip implant materials (CoCrMo alloy and Ti6Al4V) in order to decide which material is more suitable for orthopedic implants. Ti6Al4V surface manifested stable mechanical and electrochemical responses in the wear and corrosion. The test results from Ti6Al4V surface presented less coefficient of friction values and moderate change in fretting current. The experimental study concluded that the Ti6Al4V is superior to that of CoCrMo in the combined effect from mechanical loadings and electrochemical environment.

Author Response

Authors appreciate for the reviewer's comments for the manuscript. Authors made the necessary corrections and revisions based on the other reviewers' suggestions.

Reviewer 2 Report

The paper presents studies performed on Ti6Al4V and CoCrMo in a simulated synovial environment and showcased their findings such as the mechanical and corrosion behavior of these alloys. The manuscript has some interesting results and has been well written (except for some confusing lines). The manuscript can be accepted after the following changes:

  1. Please shorten the abstract
  2. Figure 1 should be made Figure 1a and add an actual picture of the experimental setup and make it Figure 1b.
  3. It would be interesting to see the SEM analysis of the specimens shown in figure 5. Please add if possible. In that way, one can know the oxide scale formed after the active sliding.
  4. Please include the active surface area of the specimen during potentiostatic measurements in figure captions (figures 13, 14, and 15)
  5. Please refer to Liu and Gilbert https://doi.org/10.1016/j.wear.2017.08.011, they have presented some interesting results on CoCrMo alloy’s fretting corrosion.

Author Response

Dear reviewer,

The necessary revision has been made based on the suggestions from reviewers. The modifications were listed and highlighted as below.

The paper presents studies performed on Ti6Al4V and CoCrMo in a simulated synovial environment and showcased their findings such as the mechanical and corrosion behavior of these alloys. The manuscript has some interesting results and has been well written (except for some confusing lines). The manuscript can be accepted after the following changes:

  1. Please shorten the abstract

Abstract was revised to highlight the concise presentation of the work.

  1. Figure 1 should be made Figure 1a and add an actual picture of the experimental setup and make it Figure 1b.

A new image of the liquid cell and specimen was added with the label in Figure 2(b).

  1. It would be interesting to see the SEM analysis of the specimens shown in figure 5. Please add if possible. In that way, one can know the oxide scale formed after the active sliding.

Unfortunately, SEM data presenting modifications of oxide layer is now available. However, micro images of the surfaces were produced and added in figure 1 to illustrate the grain sizes and phases. Chemical analysis result also was added and summarized in Table 1 on Page 3.

  1. Please include the active surface area of the specimen during potentiostatic measurements in figure captions (figures 13, 14, and 15)

Micro image of the wear tracks was used to evaluate the disturbed area by sliding cycles and included in the caption of Figures 14,15, and 16

  1. Please refer to Liu and Gilbert https://doi.org/10.1016/j.wear.2017.08.011, they have presented some interesting results on CoCrMo alloy’s fretting corrosion.

The suggested article was included in the text with a new reference number [10].
